# Orientation Normalization of Multi-Stain Skin Tissue Cross-Sections

**Ema Topolnjak**[1,2]

**Evi Paulides**[1]

**Willeke A. M. Blokx**[3]

**Mitko Veta**[1]

**Ruben T. Lucassen**[1,3]                                    R.T.LUCASSEN@UMCUTRECHT.NL

[1] *Dept. of Biomedical Engineering, Eindhoven University of Technology, the Netherlands*

[2] *Dept. of Electrical and Computer Engineering, Vanderbilt University, Nashville, TN, USA*

[2] *Dept. of Pathology, University Medical Center Utrecht, the Netherlands*

**Editors:** Accepted for publication at MIDL 2026

## Abstract

Efficient examination of skin tissue specimens is key for pathologists to keep up with an increasing workload. Normalizing the orientation of tissue cross-sections before manual assessment could contribute to a more streamlined digital workflow. In this study, we compare multiple deep learning-based approaches for predicting the rotation angle required to correct the misorientation of skin tissue cross-sections. The models were developed and evaluated using a dataset of 10,649 H&E-stained and 9,731 IHC-stained cross-section images from specimens with melanocytic lesions. Our results show that framing rotation angle prediction as a classification task with the circular target space divided into separate classes performed best, reaching mean absolute errors of 2.77° and 3.56° on the test sets of H&E and IHC-stained cross-sections, respectively, approaching the level of human annotators. Automated orientation normalization, when implemented in whole slide image viewers, could make tissue examination more efficient and convenient for pathologists, while also serving as a valuable preprocessing step for the development of position-aware or multi-stain deep learning models.

**Keywords:** Classification, Regression, Rotation Angle, Computational Pathology, Melanoma

## 1. Introduction

Whole slide imaging has enabled the transition from conventional, microscope-based tissue examination to fully digital pathology diagnostics, offering benefits such as digital archiving, remote working, as well as computational image analysis to assist pathologists (Stathonikos et al., 2013, 2020). Before a tissue specimen can be examined, however, it undergoes a histological preparation process, which includes tissue fixation, paraffin embedding, microtome sectioning, and staining. During the preparation process, tissue sections are often placed on the glass slides with an arbitrary orientation. Unlike tissue sections of most organ types, cross-sections of skin tissue specimens typically have an ideal orientation (i.e., the epidermis positioned at the top to reflect the natural outside-to-inside structure) and are, therefore, frequently misoriented, impeding optimal histological examination.

To address variation in orientation, earlier work has focused on methods such as data augmentation and rotation-equivariant convolutional neural networks (Veeling et al., 2018; Lafarge et al., 2021). While these methods improve model robustness and sample efficiency, they do not resolve the practical need of pathologists for consistently oriented tissue cross-sections. Alternatively, cross-sections can be rotated to their natural orientation prior to examination or downstream analysis. While this task has not yet been explored in dermatopathology, Shao *et al.* (Shao et al., 2024) used deep learning-based rotation angle regression for radical prostatectomy sections as preprocessing step before registration with MRI scans, and a few earlier works have investigated rotation prediction in the domain of natural images using classification (Hara et al., 2017; Gidaris et al., 2018), regression (Hara et al., 2017), and hybrid approaches (Fischer et al., 2015).

In this study, we benchmark multiple deep learning approaches for predicting the rotation angle required to correct the misorientation of skin tissue cross-sections. The models were developed and evaluated based on a dataset of 10,649 H&E-stained and 9,731 IHC-stained cross-section images from specimens with melanocytic lesions. Providing pathologists with cross-sections that are oriented consistently can streamline tissue assessment, potentially reducing the examination time and facilitating more convenient comparison across stains. Automated orientation normalization could also form an important preprocessing step in the development of position-aware or multi-stain deep learning models, as well as whole slide image (WSI) registration methods. The code and trained model parameters are made publicly available.[1]

## 2. Materials

### 2.1. Study design

This study was performed using data from the digital archive of the pathology department of the University Medical Center (UMC) Utrecht, the Netherlands. A total of 3,675 cutaneous melanocytic lesion cases with H&E-stained and IHC-stained WSIs available, accessioned between January 1, 2013, and August 31, 2023, were randomly selected with stratification. The study does not fall within the scope of the Dutch Medical Research Involving Human Subjects Act (WMO) and therefore does not require approval from an accredited medical ethics committee in the Netherlands. Nevertheless, an independent quality assessment (25U-0162) was conducted at the UMC Utrecht to ensure compliance with relevant laws and regulations, including those related to the informed consent procedure, data management, privacy, and legal considerations. Cases from patients who opted out of the use of their data for research purposes were excluded. All data were de-identified.

### 2.2. Dataset

The dataset curation process is schematically shown in Fig. A1 in Appendix A. At the start, 3,675 unique skin biopsy and excision specimens with a melanocytic lesion were identified, consisting of 175 cases for each of the 21 most frequently performed IHC stains for melanocytic lesions at the pathology department of the UMC Utrecht. To compare

---

1. https://github.com/RTLucassen/orientation_normalization

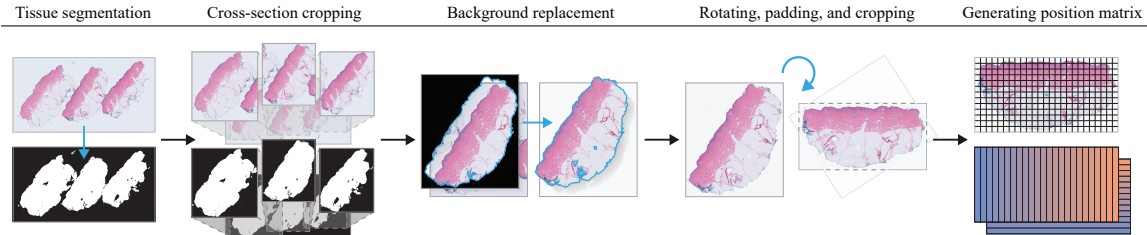

| Tissue segmentation | Cross-section cropping | Background replacement | Rotating, padding, and cropping | Generating position matrix |

Figure 1: Pipeline for image preprocessing. Tissue sections were first segmented and separated using SlideSegmenter (Lucassen et al., 2024) to crop them from the WSI at 1.25× magnification. After the background was replaced to remove artifacts and parts of adjacently positioned sections, the cross-sections were rotated to their natural orientation and the padding was corrected. Position matrices were finally generated.

the performance between stains, all available pairs of corresponding H&E-stained and IHC-stained WSIs were selected in an automated manner based on the WSI metadata.

Image acquisition was performed using a ScanScope XT scanner (Aperio, Vista, CA, USA) at 20× magnification with a resolution of 0.50 µm per pixel (226 WSIs, acquired before 2016), a NanoZoomer 2.0-XR scanner (Hamamatsu photonics, Hamamatsu, Shizuoka, Japan) at 40× magnification with a resolution of 0.23 µm per pixel (4,756 WSIs, acquired starting from 2016 until May 2022), and a NanoZoomer S360 scanner (Hamamatsu photonics, Hamamatsu, Shizuoka, Japan) at 40× magnification with a resolution of 0.23 µm per pixel (2,320 WSIs, acquired after May 2022). For the purpose of this study, all WSIs were analyzed at 1.25× magnification, as a trade-off between image detail and computational cost.

Tissue sections in the WSIs were segmented and separated using SlideSegmenter (Lucassen et al., 2024). For segmentation of the IHC-stained WSIs, the SlideSegmenter model was finetuned on an annotated set of 77 IHC-stained WSIs before use. Manual corrections to the segmentation and/or separation of the tissue sections were performed if the quality of the automated method was unsatisfactory. Moreover, we developed a custom annotation software tool for rotating the cross-sections into their natural orientation and recording the corrective rotation angle. The angle distributions for the H&E-stained and IHC-stained cross-sections on the original WSIs are shown in Fig. A2 in Appendix A. Image annotation was performed by three annotators (E.T., E.P., R.L.). In the process, sections of poor quality (e.g., no epidermis present) or sections without a natural orientation (e.g., tangentially cut tissue completely surrounded by epidermis) were excluded.

In the end, the dataset included 10,649 H&E-stained and 9,731 IHC-stained cross-section images for development and evaluation of the rotation angle prediction models. The dataset was split on a patient level into a training set (70%), a validation set (10%), and a test set (20%) for independent evaluation. Approximately one-third of the images in each set were annotated by each annotator.

| Approach | Predicted variable(s) | Loss |
|---|---|---|
| Angle regression | Angle of rotation in radians | Cosine similarity (CS) |
| Coordinate regression | Coordinate on unit circle | Cosine similarity (CS)
Mean squared error (MSE)
Mean absolute error (MAE) |
| Angle classification | Probabilities for angle classes | Categorical cross-entropy (CCE)
Binary cross-entropy (BCE) |
| Orientation classification | Probability of being correctly oriented | Binary cross-entropy (BCE) |

Table 1: Modeling objectives for orientation normalization included in the benchmark.

## 2.3. Preprocessing

The preprocessing steps are illustrated in Fig. 1. All separate cross-sections were first cropped from the original WSIs, after which the background was replaced with a uniform color equal to a representative WSI background. This was done to remove artifacts and parts of other, closely positioned cross-sections, which could affect the model predictions. The images and corresponding segmentation maps were rotated to normalize the orientation of the cross-section. Excessive padding at the edges after the rotation was removed, but also added when necessary to make the dimensions divisible by the tile size used in the deep learning model. Position matrices were generated to indicate the horizontal and vertical position of each tile with respect to the centroid of the cross-section (determined using the binary segmentation map) as the point of origin.

## 3. Methods

### 3.1. Model architecture

A Vision Transformer (ViT) (Dosovitskiy et al., 2021) (depth = 14, heads = 4, MLP-ratio = 5, embedding dimensions = 256) with 13.2 million trainable parameters was used as model architecture for all experiments unless otherwise specified. Input images were tessellated into non-overlapping tiles of $16 \times 16$ pixels and converted to feature embeddings. Because the images varied considerably in size, learnable positional embeddings, which require fixed image dimensions, were unsuitable for this application. Instead, non-learnable positional encodings were used (Vaswani et al., 2017). The positional encodings were generated based on the position matrices. Let $(p_x, p_y)$ denote the position of a tile along the horizontal and vertical axes, relative to the tile at the centroid of the cross-section as point of origin, which was assigned coordinate $(0,0)$. Both axes were encoded independently using:

$$
\mathrm{PE}(p, i) = \begin{cases} \sin\left(\dfrac{p}{10000^{\frac{2i}{d/2}}}\right) & \text{if } i \text{ is even} \\ \cos\left(\dfrac{p}{10000^{\frac{2i}{d/2}}}\right) & \text{if } i \text{ is odd} \end{cases}, \tag{1}
$$

where $p$ is the position of the tile along a single axis with respect to the origin, $i$ is the encoding dimension, and $d$ is the number of embedding dimensions used by the model. The final 2-dimensional positional encoding was obtained by concatenating the encodings for the horizontal and vertical axis. Unlike their original use in language modeling, where typically only positive coordinates are encoded, sinusoidal positional encodings can also represent negative coordinates, making them suitable for our application as well.

The parameters of the tile embedding layer, Transformer blocks, and final normalization layer were initialized based on ImageNet (Deng et al., 2009) pretrained ViT parameters from `timm` (Wightman, 2019). The final classification layer was configured with randomly initialized parameters to accommodate the difference in the prediction task.

### 3.2. Modeling objectives

We benchmarked multiple regression and classification approaches to predict the rotation angle for the orientation normalization task, which are listed in Table 1. All approaches were defined before evaluation. Starting with the regression approaches, we investigated predicting the rotation angle $\theta$ in radians, and similar to the work of Hara et al. (Hara et al., 2017), as the corresponding coordinate on the unit circle $\mathbf{v} = (\cos\theta, \sin\theta) = (x, y)$. While both formulations are suitable for optimization using the cosine similarity loss, only the latter is appropriate in combination with the mean squared error and mean absolute error loss, because optimization problems near the modulus of 360° in the circular target space are circumvented. The cosine similarity (CS) loss is defined as

$$\mathcal{L}_{\text{CS}} = \frac{1}{N} \sum_{i=1}^{N} \left(1 - \cos(\theta_i - \hat{\theta}_i)\right) = \frac{1}{N} \sum_{i=1}^{N} \left(1 - \frac{\mathbf{v}_i \cdot \hat{\mathbf{v}}_i}{|\mathbf{v}_i|\,|\hat{\mathbf{v}}_i|}\right) = \frac{1}{N} \sum_{i=1}^{N} \left(1 - \frac{x_i\,\hat{x}_i + y_i\,\hat{y}_i}{\sqrt{\hat{x}_i^2 + \hat{y}_i^2}}\right), \quad (2)$$

the mean squared error (MSE) loss is defined as

$$\mathcal{L}_{\text{MSE}} = \frac{1}{N} \sum_{i=1}^{N} \left((x_i - \hat{x}_i)^2 + (y_i - \hat{y}_i)^2\right), \quad (3)$$

and the mean absolute error (MAE) loss is defined as

$$\mathcal{L}_{\text{MAE}} = \frac{1}{N} \sum_{i=1}^{N} \left(|x_i - \hat{x}_i| + |y_i - \hat{y}_i|\right), \quad (4)$$

where $(x_i, y_i)$ and $(\hat{x}_i, \hat{y}_i)$ are the ground truth and predicted coordinates, respectively, for image $i$ in a batch of size $N$. The predicted coordinate $\hat{\mathbf{v}} = (\hat{x}, \hat{y})$ can be converted back to the predicted angle of rotation $\hat{\theta} = \text{atan2}(\hat{y}, \hat{x})$.

Continuing with the classification approaches, we investigated predicting the rotation angle class from a set of classes that divide the circular target space. The models were optimized using the categorical cross-entropy loss in combination with the softmax as final activation function and the binary cross-entropy with a sigmoid as final activation function. The categorical cross-entropy (CCE) loss was defined as

$$\mathcal{L}_{\text{CCE}} = -\frac{1}{N} \sum_{i=1}^{N} \sum_{j=1}^{C} \left(p_{ij} \log(\hat{p}_{ij})\right) \quad (5)$$

and binary cross-entropy (BCE) loss was defined as

$$\mathcal{L}_{\text{BCE}} = -\frac{1}{NC} \sum_{i=1}^{N} \sum_{j=1}^{C} \left( p_{ij} \log(\hat{p}_{ij}) + (1 - p_{ij}) \log(1 - \hat{p}_{ij}) \right), \tag{6}$$

where $p_{ij}$ and $\hat{p}_{ij}$ are the ground truth and predicted probability, respectively, for image $i$ in a batch of size $N$ and class $j$ out of the total number of classes $C$. For all multi-class classification models, the circular target space was divided into $C = 360$ angle classes. The models were optimized with label smoothing, where class labels were sampled from a wrapped normal distribution centered at the rotation angle $\theta$ with a standard deviation $\sigma = 2.5°$. The distribution was normalized for the CCE loss and unnormalized for the BCE loss. To obtain the predicted angle of rotation, the predicted probability-weighted circular mean of the center angles of each class was calculated.

Finally, we investigated a binary classification approach, predicting whether the tissue cross-section in the input image has the correct orientation. The model was optimized using the BCE loss with $C = 1$, combined with the same label smoothing as before. Model predictions across the full circular target space were obtained by rotating the input image by 1°, predicting if the orientation is correct, and repeating this 360 times. To obtain the predicted angle of rotation, the predicted probability-weighted circular mean of the angles across the repetitions was calculated.

### 3.3. Training

All models were optimized using the same training procedure. During training, randomly sampled rotations were applied to the cross-section images with normalized orientations, after which the padding was corrected again and a new position matrix was generated (see the last two steps in Fig. 1). The models were trained to predict the angle that is needed to rotate the cross-section back to its natural orientation. Based on preliminary experiments, we found the deep learning models to perform better when small rotations near the natural orientation were oversampled. Hence, the sample probability of rotation angles in the ranges of $[0°, 15°]$ and $[345°, 360°]$ was increased from $30/360 = 0.083$ to $0.25$, with a probability of 0.75 to sample from the remaining range of $[15°, 345°]$. Moreover, on-the-fly data augmentation was applied in the form of left-right flipping and color changes, including adjustments to the brightness ($\pm 0.2$), contrast ($\pm 0.2$), saturation ($\pm 0.2$), and hue ($\pm 0.05$).

The models were trained until the validation loss had converged (i.e., 500,000 iterations for regression approaches and 750,000 iterations for classification approaches) using the AdamW (Loshchilov and Hutter, 2019) optimization algorithm ($\beta_1 = 0.9$, $\beta_2 = 0.999$). Gradients were accumulated over every 20 iterations. The model parameters that resulted in the smallest loss on the validation set were saved, which was evaluated after every 5,000 iterations. The learning rate was $2 \cdot 10^{-5}$ at the start and halved after every five consecutive evaluations without a decrease in the loss on the validation set. Weight decay was equal to $1 \cdot 10^{-2}$. If the total number of tile embeddings for an image exceeded 15,000, which was selected as maximum to limit the computational resources required, a subset of the embeddings equal in size to the maximum was randomly selected. Hyperparameters were tuned based on the performance on the validation set.

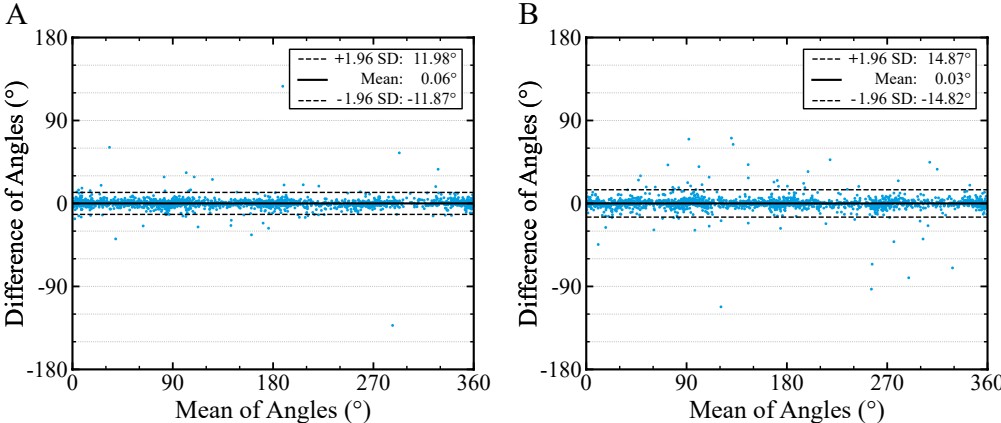

Figure 2: Bland-Altman plots of ensemble predictions from the best-performing models for the (A) H&E-stained cross-sections and (B) IHC-stained cross-sections.

### 3.4. Experimental setup

For each modeling objective in the benchmark, five model instances were trained using different random seeds. The same five seeds were consistently used across the modeling objectives. The performance of the models was evaluated based on the cross-section orientations on the original WSIs using the mean absolute error (MAE) and median absolute error (MedAE) between the ground truth and predicted rotation angle, the percentage of cases within 2.5°, 5.0°, and 10.0° of the ground truth, and Bland-Altman plots (Bland and Altman, 1986), all reported at a cross-section level. The mean and standard deviation (SD) of the performance scores for the five model instances were reported as final scores per modeling objective. All approaches were trained and evaluated on the set of H&E-stained cross-sections, the set of IHC-stained cross-sections, and the combined set. To put the model performance into perspective, all tissue cross-sections in the test set were annotated again by all three annotators (E.T., E.P., R.L.) to assess the inter-annotator variability. One of the annotators repeated the test set annotation a second time after a wash out period of two weeks to also assess the intra-annotator variability. Moreover, an ablation study was performed to show the effect of the positional encodings and the background replacement. Additional experiments and analyses are included in Appendices E, F, and G.

## 4. Results

### 4.1. Benchmark

The results for the regression and classification approaches are reported in Table 2 and Table 4 for the H&E-stained and IHC-stained cross-sections, respectively. For both staining types, angle classification using the CCE or BCE loss for training reached the best performance, followed by coordinate regression using the MSE or MAE loss. The remaining approaches, including direct angle regression in radians, coordinate regression using the CS loss, and binary orientation classification, all performed substantially worse. Similar trends were seen for a smaller ViT configuration (see Table B1 in Appendix B). Across

| Approach | Loss | MAE (°) | MedAE (°) | % of images within $k$ | | |
|---|---|---|---|---|---|---|
| | | | | $k = 2.5°$ | $k = 5.0°$ | $k = 10.0°$ |
| Angle regression | CS | $13.04 \pm 1.68$ | $7.80 \pm 0.89$ | 18.6 | 34.9 | 60.2 |
| Coordinate regression | CS | $5.71 \pm 0.49$ | $3.58 \pm 0.29$ | 36.5 | 63.9 | 87.7 |
| | MSE | $4.23 \pm 0.29$ | $2.75 \pm 0.19$ | 46.1 | 74.5 | 92.7 |
| | MAE | $3.65 \pm 0.32$ | $2.27 \pm 0.18$ | 54.1 | 80.8 | 94.3 |
| Angle classification | CCE | $2.96 \pm 0.08$ | $1.75 \pm 0.06$ | 63.7 | 86.3 | 96.4 |
| | BCE | $3.12 \pm 0.12$ | $1.82 \pm 0.05$ | 62.0 | 85.0 | 95.9 |
| Orientation classification | BCE | $5.93 \pm 0.67$ | $2.36 \pm 0.32$ | 52.3 | 74.2 | 87.7 |

Table 2: Results for the rotation angle prediction approaches, trained and evaluated on H&E-stained tissue cross-section images. The results represent the mean (and standard deviation) of five model instances per approach. Note that lower scores for the mean absolute error (MAE) and median absolute error (MedAE) are better.

all approaches, the performance was better for cross-sections stained with H&E than for cross-sections stained with IHC. Moreover, most of the models trained on the combined set of H&E-stained and IHC-stained cross-sections reached a better performance than those trained on a single staining type when evaluated on the IHC-stained cross-sections, but performed comparably to slightly worse when evaluated on the H&E-stained cross-sections (see Tables C1 and C2 in Appendix C).

For the H&E-stained cross-sections, the best performance was achieved by the angle classification models trained only on the H&E-stained cross-sections using the CCE loss, which, when averaged over five model instances, reached a MAE of $2.96 \pm 0.08°$. Further improvements were observed when using the average of the predictions to form a model ensemble instead, in which case a MAE of 2.77° was reached. The Bland-Altman plot of the model ensemble predictions is shown in Fig. 2A, demonstrating no bias and few outlier cross-sections. Visual inspection revealed the presence of separate tissue fragments and the

| Comparison | MAE (°) | MedAE (°) | % of images within $k$ | | |
|---|---|---|---|---|---|
| | | | $k = 2.5°$ | $k = 5.0°$ | $k = 10.0°$ |
| 1 vs. 1 | 1.79 | 1.03 | 80.9 | 94.1 | 98.5 |
| 1 vs. 2 | 2.18 | 1.20 | 74.7 | 91.1 | 97.3 |
| 1 vs. 3 | 2.71 | 1.55 | 69.4 | 89.3 | 96.4 |
| 2 vs. 3 | 2.64 | 1.48 | 69.2 | 88.0 | 97.4 |

Table 3: Intra-annotator variability after a 2-week period washout time (row 1) and inter-annotator variability (rows 2-4) for the H&E-stained cross-sections in the test set.

| Approach | Loss | MAE (°) | MedAE (°) | % of images within $k$ | | |
|---|---|---|---|---|---|---|
| | | | | $k = 2.5°$ | $k = 5.0°$ | $k = 10.0°$ |
| Angle regression | CS | $19.60 \pm 1.26$ | $10.73 \pm 0.95$ | 13.0 | 25.7 | 47.4 |
| Coordinate regression | CS | $9.61 \pm 0.50$ | $5.36 \pm 0.18$ | 26.3 | 47.1 | 73.3 |
| | MSE | $6.39 \pm 0.43$ | $3.74 \pm 0.16$ | 35.1 | 61.7 | 85.9 |
| | MAE | $5.87 \pm 0.49$ | $2.92 \pm 0.27$ | 44.0 | 70.2 | 88.8 |
| Angle classification | CCE | $4.46 \pm 0.21$ | $2.21 \pm 0.09$ | 55.1 | 79.0 | 92.7 |
| | BCE | $4.47 \pm 0.17$ | $2.19 \pm 0.03$ | 54.7 | 78.8 | 92.3 |
| Orientation classification | BCE | $8.50 \pm 1.12$ | $3.57 \pm 0.44$ | 38.8 | 62.3 | 80.8 |

Table 4: Results for the rotation angle prediction approaches, trained and evaluated on IHC-stained tissue cross-section images. The results represent the mean (and standard deviation) of five model instances per approach. Note that lower scores for the mean absolute error (MAE) and median absolute error (MedAE) are better.

lack of a complete epidermis among the cross-sections with the largest prediction errors, as can be seen in Fig. 3A. The inter- and intra-annotator variability for the H&E-stained cross-sections is reported in Table 3. Between the annotators, the MAE ranged from 2.18° to 2.71°. The model ensemble reached a predictive performance close to the level of consistency of the annotators. The intra-annotator comparison showed the best level of consistency.

For the IHC-stained cross-sections, the best performance was achieved by the angle classification models trained on the combined set of H&E-stained and IHC-stained cross-sections using the BCE loss, which, when averaged over five model instances, reached a MAE of $3.89 \pm 0.20°$. Similarly, further improvements were observed when using an ensemble of the five models, reaching a MAE of 3.56°. The Bland-Altman plot of the model ensemble predictions can be seen in Fig. 2B, which shows no bias, but more outliers than for the H&E-stained cross-sections. Among the IHC-stained cross-sections with the largest prediction errors, the presence of separate tissue fragments, the lack of a complete epidermis, or sections

| Comparison | MAE (°) | MedAE (°) | % of images within $k$ | | |
|---|---|---|---|---|---|
| | | | $k = 2.5°$ | $k = 5.0°$ | $k = 10.0°$ |
| 1 vs. 1 | 1.80 | 0.92 | 86.2 | 95.3 | 98.4 |
| 1 vs. 2 | 2.42 | 1.15 | 77.0 | 90.7 | 96.4 |
| 1 vs. 3 | 3.02 | 1.33 | 72.9 | 89.2 | 96.3 |
| 2 vs. 3 | 3.03 | 1.52 | 67.1 | 86.7 | 96.0 |

Table 5: Intra-annotator variability after a 2-week washout period (row 1) and inter-annotator variability (rows 2-4) for the IHC-stained cross-sections in the test set.

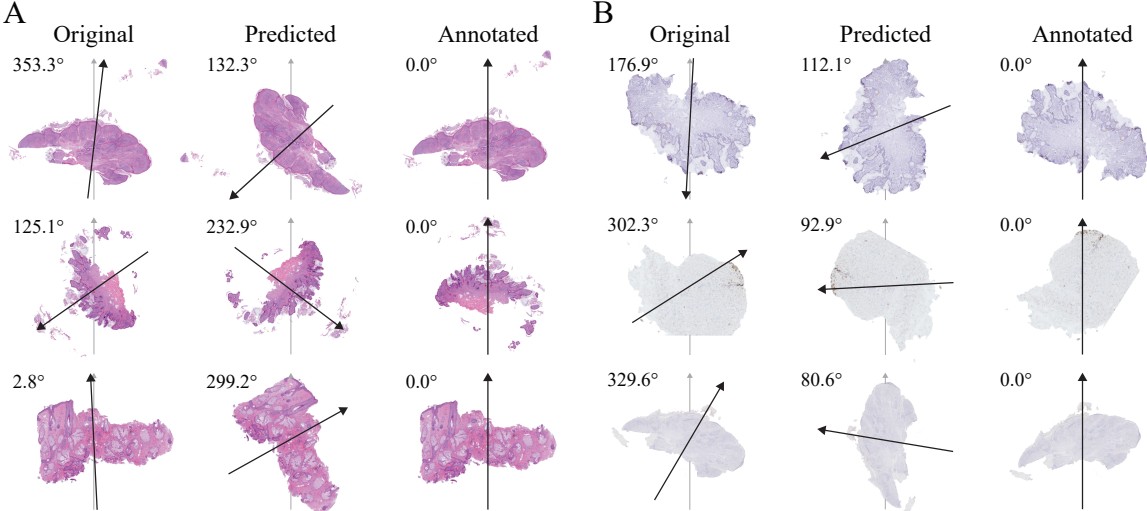

Figure 3: Three rows of (A) H&E-stained cross-sections and (B) IHC-stained cross-sections from the test set with the largest prediction errors by the ensemble of five model instances using the best-performing approach. From left to right: the original orientation on the WSI, the orientation after applying the predicted rotation angle, and the natural orientation as provided by one of the annotators.

almost completely surrounded by epidermis were primarily observed, as can be seen in Fig. 3B. The inter- and intra-annotator variability for the IHC-stained cross-sections is reported in Table 5. The MAE ranged from 2.42° to 3.03° between the annotators. A larger difference was seen between the model ensemble performance and the level of annotator consistency for the IHC-stained cross-sections than for the H&E-stained cross-sections.

### 4.2. Ablation study

The results of the ablation study are reported in Table 6. Removing the positional encodings inside the model and the background replacement during preprocessing both individually showed a decrease in the predictive performance, with a stronger decrease observed when the two were removed concurrently. Without positional encodings, the model cannot leverage any spatial relation between tiles and must rely on the patterns within the image tiles for the prediction of the rotation angle. Visual inspection of the cross-section images that showed the largest absolute errors without replaced backgrounds revealed the presence of other, closely positioned cross-sections, which would have been removed by the background replacement (see Fig. D1 in Appendix D).

## 5. Discussion and conclusion

In this work, we compared several deep learning-based classification and regression approaches for predicting the rotation angle required to normalize the orientation of H&E-

| Positional Encodings | Background replacement | | MAE (°) | MedAE (°) | % of images within $k$ | | |
|:---:|:---:|:---:|:---:|:---:|:---:|:---:|:---:|
| | Training | Evaluation | | | $k = 2.5°$ | $k = 5.0°$ | $k = 10.0°$ |
| ✗ | ✗ | ✗ | $20.79 \pm 6.27$ | $17.56 \pm 8.22$ | 9.7 | 19.2 | 36.0 |
| ✗ | ✓ | ✗ | $5.93 \pm 1.19$ | $4.10 \pm 1.02$ | 34.5 | 58.5 | 84.7 |
| ✗ | ✓ | ✓ | $5.78 \pm 1.30$ | $4.14 \pm 1.05$ | 34.0 | 58.7 | 84.8 |
| ✓ | ✗ | ✗ | $5.39 \pm 0.37$ | $3.31 \pm 0.32$ | 40.3 | 66.7 | 88.7 |
| ✓ | ✓ | ✗ | $3.45 \pm 0.10$ | $1.90 \pm 0.09$ | 60.5 | 83.2 | 94.7 |
| ✓ | ✓ | ✓ | $2.96 \pm 0.08$ | $1.75 \pm 0.06$ | 63.7 | 86.3 | 96.4 |

Table 6: Results of ablation study showing how the predictive performance is affected by removing the positional encodings and background replacement during preprocessing on the H&E-stained cross-section images. The results represent the mean (and standard deviation) of five model instances per approach. Note that lower scores for the mean absolute error (MAE) and median absolute error (MedAE) are better.

stained and IHC-stained skin tissue cross-sections. Among the evaluated approaches, angle classification in combination with the BCE or CCE loss for training reached the best performance.

For all evaluated approaches, the performance was better for cross-sections stained with H&E than for cross-sections stained with IHC, which might be due to the higher contrast of the tissue with respect to the background or because of better tissue quality. This differs from the level of consistency between the human annotators, which was similar for both staining types. In addition, training on the combined set of H&E and IHC-stained cross-sections, compared to training exclusively on cross-sections from a single staining type, resulted for most approaches in slightly worse performance when evaluated on the test set of H&E-stained cross-sections, while mostly reaching a better performance on the test set of IHC-stained cross-sections.

An interesting direction for future work would be to integrate orientation normalization of skin tissue cross-sections into a WSI viewer, followed by a user study to assess the effect on the diagnostic workflow (e.g., reduction in examination time or improvements in the convenience for pathologists). Automated matching of corresponding cross-sections across stains after the orientation has been normalized can potentially also be helpful. Moreover, the predictive performance on skin tissue cross-sections with non-melanocytic pathologies remains to be evaluated.

In conclusion, the best performance was achieved by approaching rotation angle prediction as a classification task with the circular target space divided into separate classes, which reached a performance close to the consistency level of human annotators. Automated orientation normalization could both streamline skin tissue assessment by pathologists, potentially reducing the examination time and improving convenience, and form an important preprocessing step for developing position-aware or multi-stain deep learning models.

## Acknowledgments

This research was financially supported by the Hanarth Fonds.

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

# Appendix A. Dataset details

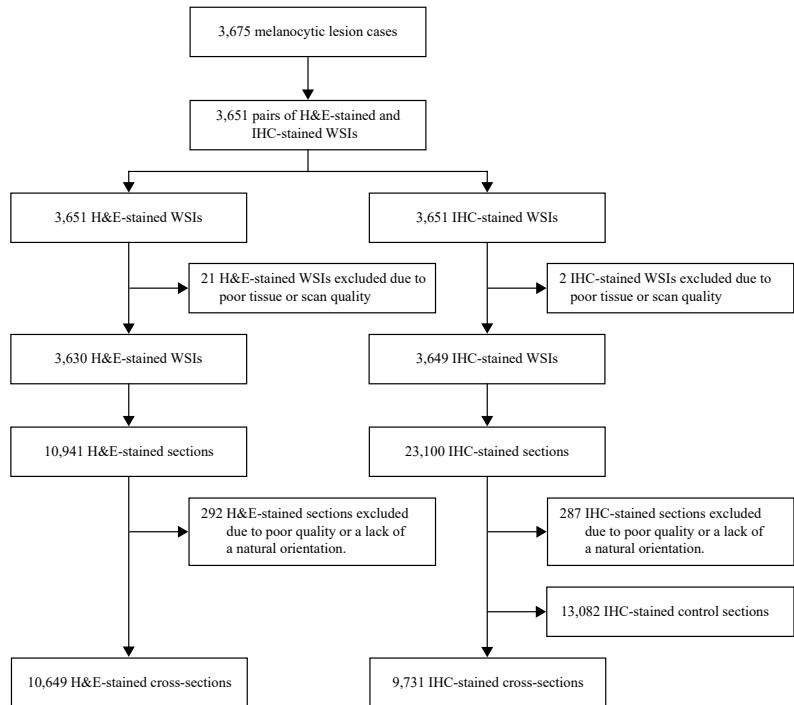

Figure A1: Flow chart of the dataset curation process.

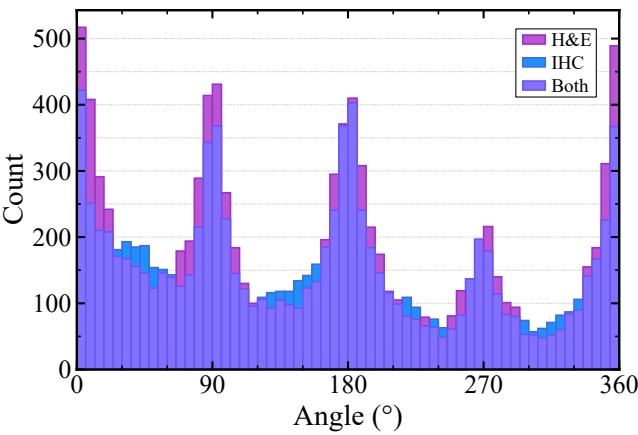

Figure A2: Histogram showing the distribution of angles for the H&E-stained and IHC-stained cross-sections on the original WSIs. The distributions are similar for the different staining types, both showing peaks at 0°/360°, 90°, 180°, and 270°, indicating that the lab technicians did to some extent take the orientation into account when placing the tissue sections on the microscopy slides.

## Appendix B. Results for smaller ViT configuration trained on H&E-stained cross-sections

| Approach | Loss | MAE (°) | MedAE (°) | % of images within $k$ | | |
|---|---|---|---|---|---|---|
| | | | | $k = 2.5°$ | $k = 5.0°$ | $k = 10.0°$ |
| Angle regression | CS | $19.10 \pm 2.04$ | $11.81 \pm 1.50$ | 12.3 | 23.8 | 44.0 |
| Coordinate regression | CS | $11.08 \pm 0.76$ | $6.53 \pm 0.41$ | 20.8 | 40.5 | 66.8 |
| | MSE | $8.01 \pm 0.23$ | $4.66 \pm 0.16$ | 29.9 | 52.6 | 79.0 |
| | MAE | $7.37 \pm 0.52$ | $3.87 \pm 0.15$ | 35.0 | 60.0 | 83.4 |
| Angle classification | CCE | $5.60 \pm 0.27$ | $2.83 \pm 0.08$ | 45.3 | 72.2 | 90.1 |
| | BCE | $6.05 \pm 0.23$ | $3.00 \pm 0.12$ | 43.4 | 69.5 | 88.6 |
| Orientation classification | BCE | $30.20 \pm 1.32$ | $11.03 \pm 1.08$ | 16.0 | 29.6 | 47.7 |

Table B1: Results for the rotation angle prediction approaches, trained and evaluated on H&E-stained tissue cross-section images using a smaller ViT configuration (depth = 8, heads = 4, MLP-ratio = 4, embedding dimensions = 128) with 1.7 million trainable parameters. Note that lower scores for the mean absolute error (MAE) and median absolute error (MedAE) are better.

## Appendix C. Results for models trained on combined set of H&E and IHC-stained cross-sections

| Approach | Loss | MAE (°) | MedAE (°) | % of images within $k$ | | |
| --- | --- | --- | --- | --- | --- | --- |
| | | | | $k = 2.5°$ | $k = 5.0°$ | $k = 10.0°$ |
| Angle regression | CS | $15.46 \pm 2.22$ | $9.26 \pm 1.20$ | 15.8 | 29.4 | 53.1 |
| Coordinate regression | CS | $7.40 \pm 0.49$ | $4.10 \pm 0.19$ | 32.2 | 58.0 | 82.7 |
| | MSE | $4.61 \pm 0.30$ | $2.86 \pm 0.16$ | 44.8 | 72.7 | 91.5 |
| | MAE | $4.08 \pm 0.59$ | $2.33 \pm 0.22$ | 53.2 | 79.4 | 93.4 |
| Angle classification | CCE | $3.49 \pm 0.17$ | $1.89 \pm 0.09$ | 60.6 | 83.4 | 94.9 |
| | BCE | $3.22 \pm 0.21$ | $1.82 \pm 0.08$ | 62.5 | 85.1 | 95.7 |
| Orientation classification | BCE | $6.92 \pm 0.36$ | $2.49 \pm 0.09$ | 50.4 | 72.3 | 86.5 |

Table C1: Results for the rotation angle prediction approaches, trained on the combined set of H&E-stained and IHC-stained cross-sections images, evaluated only on the H&E-stained tissue cross-section images. The results represent the mean (and standard deviation) of five model instances per approach. Note that lower scores for the mean absolute error (MAE) and median absolute error (MedAE) are better.

| Approach | Loss | MAE (°) | MedAE (°) | % of images within $k$ | | |
| --- | --- | --- | --- | --- | --- | --- |
| | | | | $k = 2.5°$ | $k = 5.0°$ | $k = 10.0°$ |
| Angle regression | CS | $20.66 \pm 2.89$ | $11.96 \pm 1.96$ | 12.3 | 24.2 | 43.9 |
| Coordinate regression | CS | $9.34 \pm 0.93$ | $5.24 \pm 0.30$ | 26.2 | 47.9 | 74.9 |
| | MSE | $5.85 \pm 0.37$ | $3.57 \pm 0.13$ | 37.2 | 63.1 | 87.0 |
| | MAE | $5.20 \pm 0.67$ | $2.77 \pm 0.18$ | 46.0 | 72.6 | 90.2 |
| Angle classification | CCE | $4.31 \pm 0.27$ | $2.18 \pm 0.08$ | 55.4 | 79.6 | 92.5 |
| | BCE | $3.89 \pm 0.20$ | $2.08 \pm 0.10$ | 57.3 | 81.1 | 93.8 |
| Orientation classification | BCE | $7.91 \pm 0.61$ | $3.11 \pm 0.10$ | 42.8 | 65.0 | 82.1 |

Table C2: Results for the rotation angle prediction approaches, trained on the combined set of H&E-stained and IHC-stained cross-sections images, evaluated only on the IHC-stained tissue cross-section images. The results represent the mean (and standard deviation) of five model instances per approach. Note that lower scores for the mean absolute error (MAE) and median absolute error (MedAE) are better.

## Appendix D. Visual examples showing the benefit of background replacement

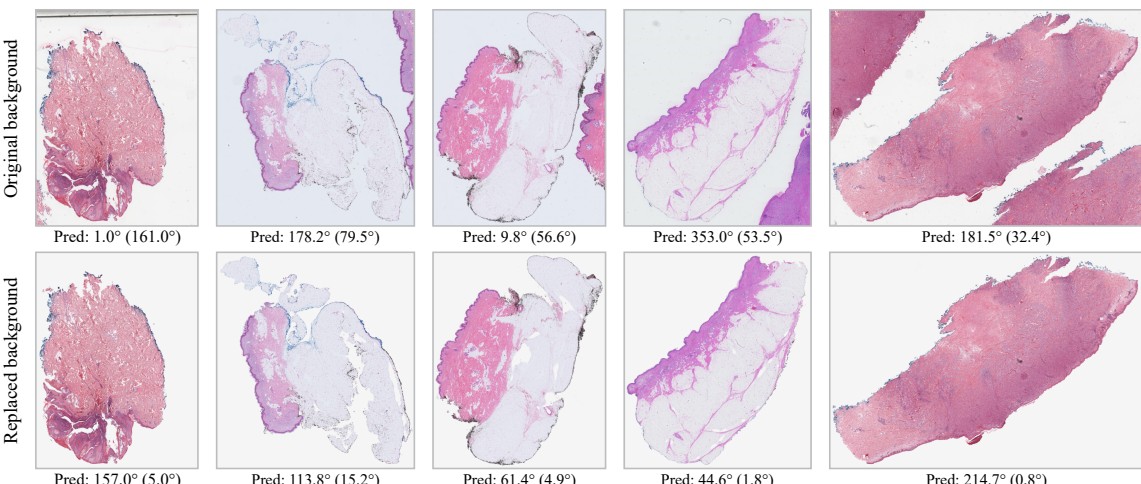

Figure D1: Five H&E-stained cross-section images from the test set with the original background in the top row and the replaced background in the bottom row. The predicted rotation angle is shown below each image, with the absolute error relative to the ground truth in parenthesis. The predictions were obtained using angle classification with a single ViT that was trained on cross-section images with replaced backgrounds.

## Appendix E. Inference time and GPU memory consumption

To integrate orientation normalization of skin tissue cross-sections into WSI viewers, the inference time and GPU memory consumption of the rotation angle prediction model are important to consider. While the optimization of these metrics was not the focus of this study, we do report statistics of the inference time and GPU memory consumption measurements for the best-performing ViT on the test set in Table E1 and E2. The measurements are reported at several percentiles because the cross-section images vary in size. Limiting the number of feature embeddings to a maximum of 15,000 reduced the peak inference time and GPU memory consumption substantially. The effect of sampling a subset of the feature embeddings on the predictive performance is investigated in Appendix F.

| Percentile | 50 Median | 75 | 95 | 99 | 100 Max |
|---|---|---|---|---|---|
| Embeddings | 2,068 | 4,285 | 9,696 | 16,190 | 50,629 |
| Time (ms) | 9.3 | 17.4 | 62.3 | 156.6 | 2,022.6 |
| Memory (GB) | 0.3 | 0.7 | 3.1 | 8.2 | 78.7 |

Table E1: Inference time and GPU memory consumption at selected percentiles across the H&E-stained test set using a single ViT model for angle classification, evaluated on an NVIDIA A100 80 GB GPU with no limit on the number of feature embeddings per image.

| Percentile | 50 Median | 75 | 95 | 99 | 100 Max |
|---|---|---|---|---|---|
| Embeddings | 2,068 | 4,285 | 9,696 | 15,000 | 15,000 |
| Time (ms) | 7.2 | 8.8 | 14.2 | 27.5 | 53.5 |
| Memory (GB) | 0.3 | 0.7 | 3.1 | 7.1 | 7.2 |

Table E2: Inference time and GPU memory consumption at selected percentiles across the H&E-stained test set using a single ViT model for angle classification, evaluated on one-quarter of an NVIDIA A100 80 GB GPU with the maximum number of feature embeddings per image limited to 15,000.

## Appendix F. Effect of feature embedding sampling on the predictive performance

Sampling a subset of the feature embeddings extracted from an image can be effective to reduce the inference time and GPU memory consumption. In Figure F1, the predictive performance as a function of the percentage of embeddings sampled to make the prediction is shown. The performance decreased slightly from using 100% to 10% of the feature embeddings, and strongly when less than 10% of the feature embeddings were sampled per image.

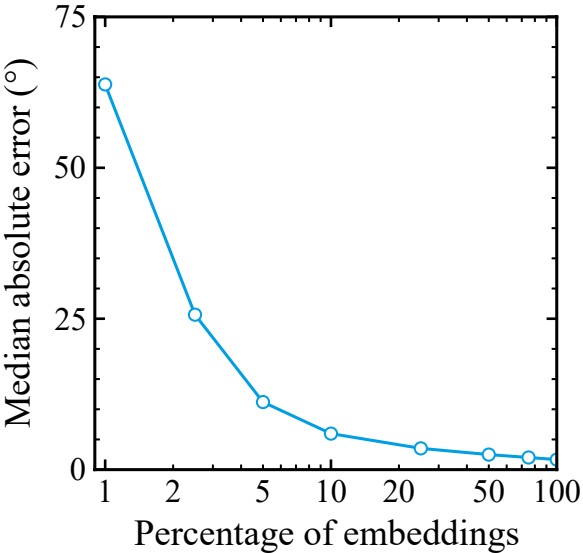

Figure F1: Median absolute error (MedAE) on the H&E-stained test set using angle classification with a single ViT based on randomly sampled subsets of the feature embeddings for each image. All data points represent the average across 25 repetitions. Note that logarithmic scaling is used for the x-axis.

## Appendix G. Uncertainty estimation based on the predicted probability distribution

Estimating the uncertainty of model predictions can potentially be an effective approach to identify incorrect predictions or detect unsuitable input images. In Figure G1, the entropy of the predicted probability distribution is plotted against the absolute error between the predicted and ground truth rotation angle. A moderate correlation is observed between the absolute error and the entropy for the H&E-stained test set images (Pearson's correlation coefficient $r = 0.58$ and Spearman's correlation coefficient $\rho = 0.41$), and the predictions for approximately half of the images excluded during preprocessing because of poor tissue quality or a lack of a natural orientation have a comparatively high entropy. Hence, while not perfect, the entropy of the predicted probability distribution could be helpful for identifying incorrect predictions and detecting unsuitable input images. An alternative approach, which could be explored in future work, would be to add an auxiliary classifier to predict whether a natural orientation can be defined for the cross-section as quality assurance.

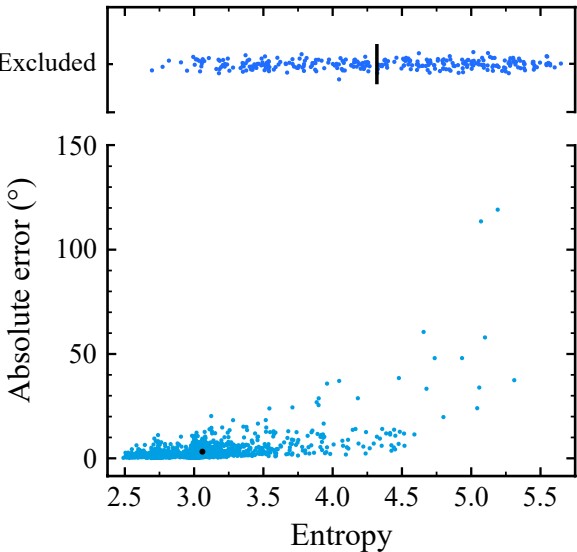

Figure G1: Absolute error between the predicted and ground truth rotation angle visualized against the entropy of the predicted probability distribution for all H&E-stained cross-section images in the test set using the angle classification method with an ensemble of five ViT models. At the top, the entropy of the predicted probability distribution using the same model ensemble is shown for all H&E-stained cross-section images that were excluded during the dataset annotation stage because of poor tissue quality or a lack of a natural orientation, which have no ground truth rotation angle or absolute error for this reason. The black dot and vertical bar represent the mean for the test set and excluded images, respectively.

