# OpenReview forum: "Orientation Normalization of Multi-Stain Skin Tissue Cross-Sections"
_MIDL.io/2026/Conference — MIDL 2026 Poster_

### Official Review · Reviewer_viau · 2026-01-09

**Confidence:** 4
**Preliminary Rating:** 4
**Final Rating:** 4

**Summary:**

In this paper, the authors address the problem of dermatology slides not being optimally rotated for efficient examination. They formulate the problem as different tasks (e.g. rotation angle regression, rotation angle class classification) and train and test a ViT model for these tasks. They show that formulating the problem as a rotation angle class classification yields best results for both H&E and IHC stained slides and that the rotation error is similar to human inter-observer variance.

**Strengths:**

* The paper is very clearly structured and well written. It is easy to follow
* The experiments carried out are suitable for testing the methods, and the conclusions follow logically from the results.
* Human observers were included and inter- and intra-observer variance was measured
* 3 different scanners were used for dataset generation

**Weaknesses:**

* Clinical motivation could be stronger. Efficiency is the most prominent argument here but it is not shown how much gain there is in practice, i.e. how long does it take a human to rotate the slide before examination? (The data is maybe even present from performing the annotations). It seems this approach cuts 1-2 seconds from pathologists workload (which is probably the same net amount of time needed to perform the automatic rotation).
* An obvious alternative approach would be to use AI to detect the epidermis regions in the image and rotate the slide such that this region is pointing to the top. This would probably be feasible with a small segmentation or tile classification model. A comparison to such an approach would be beneficial

**Detailed Comments:**

* Using a ViT seems to be overkill for this task. It would be interesting to see how smaller models perform at this.
* Where do the described peaks at multiples of 90° stem from? This should be explained and not just stated in the figure legend. Is it an artifact of the annotation tool as it maybe has 90° rotation buttons? This would introduce an unwanted bias.
* On-the-fly augmentation includes vertical flipping. Doesn't this undermine the oversampling of near-natural orientations?
* While the Bland-Altman plots are really useful, a smaller y-axis would have helped (-90 to 90°) to better interpret the results.
* The result tables have percentages of cases below certain error thresholds. What is the rationale behind these specific numbers? Do you have measurements or at least educated guesses how much rotation pathologists tolerate (before getting slower)?
* There is quite a number of methods/models being tested and compared on the test set (7 losses times ensemble vs single). It would be good to state that the compared methods were all defined before testing so that readers do not suspect multiple testing

**Justification Of Final Rating:**

The authors addressed most of my concerns in the discussion and the manuscript. The remaining issue is the motivation of the paper / problem analysis, which is still not very strong. It would have been beneficial to substantiate the general problem statement with some data.

**Justification Of The Preliminary Rating:**

The paper presents a problem that has not really been addressed yet and show that it can be solved with the proposed approaches. However, the importance of this problem appears to be rather small and alternative ways of approaching it have not been given.

**Questions To Address In The Rebuttal:**

See weaknesses and detailed comments

---

> ### Author Response · Authors · 2026-01-25
>
> We thank the reviewer for providing feedback on our manuscript. Our response to the suggestions and raised questions are listed below:
>
> Weaknesses:
> 1. We understand the concerns of the reviewer regarding the clinical motivation. While saving 1-2 seconds for normalizing the orientation of a cross-section by a pathologist may not seem worthwhile, it is important to remark that there are often multiple cross-sections on a single slide, and there are usually multiple H&E-stained slides per specimen. In addition, especially for more challenging cases, up to 10 IHC-stained slides are regularly prepared.  Hence, we expect that the time gain can be considerably larger than a few seconds for most cases. Considering that the largest volume of cases tend to be skin specimens for a typical pathology lab, the accumulated reduction could be meaningful. Moreover, there is often quite a lot of time between the acquisition of the whole slide image (WSI) and the moment when a pathologist starts to examine the WSI (e.g., scanning can be done overnight). In a fully digital pathology lab, this provides the opportunity to perform the segmentation and orientation normalization directly after acquisition, such that the model predictions will already be available when the pathologists examine the tissue. We did not record the time it took the annotators to rotate the cross-section images during the annotation process, which in hindsight would have been interesting to collect as well. It could likely also increase the convenience for the pathologist by reducing the mouse and keyboard interactions and make the comparison of corresponding H&E and IHC-stained cross-sections easier. Conducting a user study to investigate if orientation normalization benefits dermatopathologists during routine tissue examination would be interesting for future work, which we have included in the discussion section (p. 11). Next to the clinical motivation, there is of course also the application as preprocessing tool for downstream deep learning models of WSI registration methods, which may benefit from cross-sections with a normalized orientation.
> 2. We share the opinion that rotating a section based on a segmentation of the epidermis will likely also work. The reason why we did not opt for this approach is that annotating the epidermis is considerably more time-consuming than rotating a section, and we are unaware of any publicly available models that can perform epidermis segmentation at low resolution (and performing this task at high resolution is likely too slow in practice). Furthermore, we believe that this approach can more easily be adapted to tissue sections of other organs, albeit with a different definition of natural orientation, in comparison to methods that rely on the segmentation of one specific structure.

---

> > ### Author Response · Authors · 2026-01-25
> >
> > Detailed Comments:
> > 1. We selected a Vision Transformer (ViT) for the experiments because it is currently one of the most frequently used and empirically most successful deep learning model architectures. The ViT configuration we used is much smaller (13.2 million trainable parameters) than the configurations in the original ViT paper (86, 307, and 632 million).  One potential advantage of using a ViT in comparison to a convolutional neural network is the self-attention mechanism, which allows the model to learn long-range associations (e.g., tiles from opposite edges of a cross-section) already early on in the network, which could be advantageous for this task (e.g. contrasting the location of the epidermis with the deep stromal tissue). We also benchmarked a smaller ViT with 1.7 million parameters (please see Appendix B), which showed similar trends, but a lower performance across all approaches, indicating that the model size we initially used is at least to some extent beneficial.
> > 2. The histogram shows the orientations of the cross-sections on the original WSIs. It can be seen that all orientations are present, but there are peaks at 0/360°, 90°, 180°, and 270°. This suggests that the lab technicians who place the tissue cross-sections on the microscopy slides are sometimes able to take the orientation into account. We included this in the caption of the histogram (Figure A2).
> > 3. For clarification, we used left-right flipping on the normalized images to augment the dataset with mirror images without any changes being required to the labels. We corrected this in the text (p. 6).
> > 4. We appreciate the suggestion. There are a small number of points, however, that fall outside of the [-90°, 90°] range. While it would make the interpretation around the mean slightly easier with the y-axis limited to this range, we prefer to show all data points, including the largest failure cases.
> > 5. The lowest value (k = 2.5°) was selected to be near the inter-annotator difference. While we cannot say for certain at what angle misorientations start to impede pathologists in their examination (and this may very well differ per person), this will be beyond the expected inter-observer agreement of around 2.5°, which is why we also wanted to show the performance at larger values of k. In a future user study, it would be good to determine at what level of misorientation pathologists generally are impeded.
> > 6. We included the statement that all compared methods were defined before testing (p. 5).

---

> ### Comment · Reviewer_viau · 2026-01-28
>
> Thank you for the responses.
> Reg the weaknesses: I see that manual correction of orientation can add up to non-neglectable amounts of time.It would still be interesting what share of the total examination time this makes up.
> Regarding the multiple sections per slide, in case they are all more or less randomly distributed on the slide, you automated rotation might even be part of a safety feature because pathologists might get confused which section they already examined when rotating the slide a lot. In this case, this would strengthen the (otherwise still not particularly strong) motivation of the paper.
>
> Regarding the comments: Thank you for the clarifications and corrections.

---

### Official Review · Reviewer_pkiQ · 2026-01-09

**Confidence:** 4
**Preliminary Rating:** 3
**Final Rating:** 4

**Summary:**

The authors test several objectives: angle regression, coordinate regression, angle classification, and orientation classification. They propose circular regression/classification with label smoothing to account for angle wrapping. Experiments use a large, well-curated dataset and include inter- and intra-annotator comparisons. Results show some improvement in predicting rotation angles, but practical impact and broader comparisons are limited.

**Strengths:**

Large and well curated dataset with multiple stains.
While not novel, good justification and design of circular regression/classification, and label smoothing with wrapped distributions.
Includes inter-annotator and intra-annotator comparisons, giving context to human-level variability.

**Weaknesses:**

No comparison with other existing methods (see below).
Limited novelty; most ideas are known, though design choices are reasonable.
Model is large and expensive to train (500k–750k iterations).
Error analysis is shallow (see below).
Pretraining on ImageNet may not be optimal for histopathology due to rotation invariance.

**Detailed Comments:**

1) No comparison with other methods. only the ViT with different targets. It could be compared with the methods mentioned in introduction.
2) The model seems complex, large and train for long (500k/750k iterations),
3) Error analysis could be deeper: no stratified analysis by tissue size, epidermis completeness, stain type, or number of fragments.
4) Clinical impact: Benefit for pathologists is not analyzed with a user study. And it is not clear what level of angular error is really acceptable in practice.
5) The model is pretrained on imagenet. There are many FMs now for histopathology. Also pretrained on ImageNet requires a strong invariance to rotation, which is not optimal for the task.
6) “The models were trained until the validation loss had converged” clarify which early stopping/patience was used.
7) If I understand correctly you mean increased from 30/360=0.083 to 0.25? Clarify in the text. “Hence, the sample probability of rotation angles in the ranges of [0°, 15°] and [345°, 360°] was increased to 0.25, with a probability of 0.75 to sample from the remaining range of [15°, 345°].”
8) Typos: “and the was padding corrected.”
9) Potentially missing related work:
Shao, Wei, et al. "RAPHIA: A deep learning pipeline for the registration of MRI and whole-mount histopathology images of the prostate." Computers in biology and medicine 173 (2024): 108318.

**Justification Of Final Rating:**

The authors clarified some of my points and added some additional analyses. While the novelty and depth are still limited, the results are interested and it can be sufficient for a MIDL paper. I change my rating to weak accept.

**Justification Of The Preliminary Rating:**

The paper seems technically sound and uses a solid dataset, but its novelty is limited and direct comparisons to other methods are missing. Error analysis is shallow and practical impact remains weak.

**Questions To Address In The Rebuttal:**

Extend error analysis as discussed above.
Inference-time efficiency: Compare runtimes and discuss feasibility for integration into WSI viewers
Clarify acceptable error margins.

---

> ### Author Response · Authors · 2026-01-25
>
> We thank the reviewer for reviewing our manuscript and providing several directions for improvements. Our response to the suggestions and raised questions are listed below:
>
> Weaknesses:
>
> Most of the weaknesses that were pointed out by the reviewer are discussed in more detail below. We consider the main novelty of our work to be the application, as to the best of our knowledge, no other studies have previously developed orientation normalization methods for skin tissue cross-sections in histopathology, nor have any benchmarked this many approaches for rotation angle prediction.
>
> Detailed Comments (part 1):
> 1. The aim of our paper was to investigate what modeling objective is most suitable for the task of rotation prediction in histopathology. As mentioned in the introduction, several methods have been proposed for similar tasks using natural images. We decided to compare these methods (which also includes the modeling objective used in the suggested related work at point 9 of the Detailed Comments) using the same Vision Transformer (ViT) architecture to exclude the influence of differences between model architectures. Based on the request from the other reviewers, we have repeated the benchmark using a smaller ViT configuration to investigate if the findings generalize (please see Appendix B). Similar trends were observed using the smaller ViT. We also attempted to train a ResNet18 and EfficientNet-B3, but this resulted in training instability due to the batch normalization operation which is incompatible with our training strategy (i.e., a batch size of 1 with gradient accumulation to allow for cross-section images with large differences in size). We considered the Swin transformer as well, but the official implementation does not support training with images of different sizes. Furthermore, the methods by (Veeling et al., 2018) and (Lafarge et al., 2021) mentioned in the introduction focus on downstream tasks, for example tumor classification, by learning rotation invariant representations. This is an alternative approach to normalizing the orientation before downstream analysis. These methods are, therefore, unsuitable for the task of predicting the orientation itself.
> 2. We used a standard implementation of the ViT architecture, with a configuration that resulted in 13.2 million trainable parameters. For clarification, we would like to point out that all models were trained with a batch size of 1 using gradient accumulation to address the problem of large differences in image size between the cross-sections. Gradients were accumulated over 20 iterations, meaning that the network parameters were updated 25,000 or 37,500 times. Considering that the dataset size is approximately ~7,500 cross-sections, this equates to roughly 67 and 100 epochs. The training time was at most ~38 hours on a single GPU. In our opinion, the model is not extremely complex, large, or demanding to train. We have also repeated the benchmark with a smaller ViT configuration of 1.7 million parameters and saw the same trends, but lower performance across all methods, showing that the larger ViT configuration is beneficial. Moreover, we have also made all model parameters available such that other researchers can benefit from this work without having to repeat any training.

---

> > ### Author Response · Authors · 2026-01-25
> >
> > Detailed Comments (part 2):
> >
> > 3. We share the opinion of the reviewer that further evaluation of the performance could be valuable. We further investigated the correlation between the absolute error and the size of the cross-section, as well as the number of fragments, but did not see strong correlations (please see the supporting document). During the annotation stage, we excluded all cross-sections for which no natural orientation could be assigned. We did not record whether the epidermis is complete or incomplete for the included cross-sections. In addition, we are unaware of any epidermis segmentation models at low magnification, preventing us to investigate the association between the epidermis length and the absolute error further. A lack of a complete epidermis was, however, noticeable during visual inspection of the largest failure cases, which can also be seen in figure 3A (third row) and 3B (second row). We already report the results separately for the H&E and IHC-stained cross-sections in the manuscript. While differences were seen when subdividing the dataset further based on the specific IHC staining and the corresponding H&E-stained whole slide images (WSIs) (please see the supporting document), identifying the underlying causes remains challenging, as these differences could also be driven by the random sampling of cases when the dataset was collected.
> > 4. We agree with the reviewer that it would be very interesting to investigate in a user study what the benefit of the orientation normalization could be. However, seamlessly integrating our model into the WSI viewer that is currently used in our pathology lab requires additional work. We included a sentence in the discussion section that a user study would be an important direction for future work (p. 11). Moreover, investigating at which degree of misorientation a pathologist is impeded in their examination could also be part of a future user study.
> > 5. We used ImageNet pretrained parameters as these may provide a better starting point for optimization than randomly initialized weights, possibly leading to faster convergence. All parameters were trainable and could be updated by the model, so we do not expect that this will have a large effect on the performance of the model after training has been completed. The reviewer is correct that there are many foundation models for histopathology nowadays, however, these are all trained on high magnification tiles (20x or 40x). We deliberately decided to develop our models using low magnification images, as the most important structures for the orientation (e.g., the epidermis) are still visible, while the computational cost is reduced substantially.
> > 6. We defined the total number of iterations for training based on several preliminary training runs. During training, no early stopping or patience was used. Instead, the model parameters that resulted in the best performance on the validation set during intermediate evaluations were selected as the final model parameters. We did use a patience parameter for reducing the learning rate during training.
> > 7. The assumption is correct, we indeed increased the sample probability from 30/360=0.083 to 25. We added this in the manuscript to make it more clear (p. 6).
> > 8. We corrected the typo in the manuscript.
> > 9. Thank you for bringing this study to our attention. We have included it in the introduction as part of the related works (p. 2).

---

> > > ### Comment · Reviewer_pkiQ · 2026-01-27
> > >
> > > Thank you for the clarifications and additional analyses. Based on this, I will change my rating to weak accept.
> > > 1. Good to have tested a smaller ViT
> > > 2. Thank you for the clarification. It is indeed not extremely complex.
> > > 3. ok
> > > 4. It is an important missing information/analysis but not strictly necessary for a MIDL paper.
> > > 5. Some FMs are multi magnification, eg Virchow2. I still think that starting from ImageNet pretrained weights is not an optimal choice because of the rotation invariance (and domain shift), but I agree that it shouldn’t harm too much and the rotation equivariance should be easily re-learned.
> > > 6. ok, it is no big deal, but you could have stopped on validation loss since it seems to be what you do manually in this sentence.
> > > 7-8-9. ok

---

### Official Review · Reviewer_Vmty · 2026-01-10

**Confidence:** 4
**Preliminary Rating:** 4

**Summary:**

This paper addresses a practical digital pathology workflow problem: skin tissue cross‑sections are frequently placed on slides with arbitrary rotation, while pathologists typically prefer a consistent “natural” orientation (epidermis at the top). The authors propose and benchmark several deep learning formulations to predict the corrective rotation angle for skin cross‑sections. They curate a large dataset from UMC Utrecht consisting of 10k H&E and 9k IHC cross‑section images from cutaneous melanocytic lesion specimens, spanning multiple scanners and years. Cross‑sections are extracted via segmentation/cropping (SlideSegmenter), background replacement to remove neighboring section artifacts, and annotation using a custom tool by three annotators; low-quality and inherently ambiguous cases (e.g., no epidermis; tangential cuts surrounded by epidermis) are excluded. The dataset is split at patient level in to train/validation/test (70/10/20) sets.

All methods share a Vision Transformer backbone (ViT; depth 14, 4 heads, embed dim 256) operating on 16×16 tiles with a cap of 15,000 tokens; variable-sized inputs are handled using sinusoidal positional encodings derived from per‑tile position matrices relative to the tissue centroid. The benchmark compares (1) angle regression, (2) unit‑circle coordinate regression (sin/cos) with multiple losses, (3) 360‑class angle classification (CCE or BCE) with wrapped label smoothing, and (4) a computationally expensive orientation classifier scanned over 360 rotations.

Results show that 360‑class angle classification performs best. Using ensembles, the reported test performance reaches MAE 2.77° (H&E) and 3.56° (IHC), with Bland–Altman analysis indicating little bias and most failures associated with fragmented tissue or incomplete epidermis. Human re‑annotation of the test set provides inter-/intra‑annotator variability: the H&E ensemble is close to inter‑annotator consistency, while the IHC gap is larger.

**Strengths:**

Well-motivated workflow contribution for dermatopathology. Orientation normalization is a clear usability improvement for slide navigation and cross‑stain comparison, and the manuscript explicitly connects this to viewer integration and downstream modeling.

Large, multi‑stain dataset and careful annotation design. The dataset size (10k+ H&E, 9k+ IHC), multi‑scanner acquisition, and the inclusion of inter-/intra‑annotator variability analyses strengthen the empirical credibility.

Clean benchmark focused on objective design in a circular target space. The comparison between angle regression, unit‑circle regression, and circular classification is structured and instructive with meaningful wrapped label smoothing and circular mean decoding.

Strong performance with qualitative and agreement analyses. Reporting ensembles, Bland–Altman plots, and visual failure cases provides meaningful insight beyond a single metric.

Practical handling of variable-size histology crops. The centroid-relative position matrix + sinusoidal encoding is a reasonable design choice when images vary widely in spatial extent.

**Weaknesses:**

1. Architecture/baseline coverage: objective conclusions are only demonstrated with one backbone family. The main conclusion—“360‑class angle classification > regression for rotation prediction”—is currently shown only for a single ViT configuration. This makes it hard to know whether the gain is fundamentally due to the objective/target representation or partly tied to the specific ViT setup (tiling, positional encoding, token subsampling). One additional backbone baseline (such as group or rotation equivariant CNN or swin transformers, or a smaller ViT variant etc.) whose purpose is not to beat ViT, but to test whether the objective ranking is architecture‑agnostic and to better characterize compute/latency tradeoffs is required.

2. Scope/generalization: the paper is skin-specific and even within skin the “natural orientation” is not always defined. The dataset is restricted to melanocytic lesion specimens; moreover, the authors explicitly exclude cases with no epidermis and cases without a natural orientation (e.g., tangential cuts surrounded by epidermis). The largest-error examples also frequently involve fragmented tissue or incomplete epidermis. This makes the method’s behavior in real deployment (where such cases occur) insufficiently characterized. Some of the additional assessments needed are:

- Within dermatopathology: evaluate or at least stratify performance across additional, common skin specimen contexts where epidermis may be partial/fragmented in different ways (e.g., inflammatory dermatoses, BCC/SCC excisions, heavily ulcerated lesions, shave biopsies), or provide a failure‑mode breakdown using morphology proxies (fragmentation count, epidermis length fraction, etc.).

- Abstention/quality gating: train an auxiliary classifier to predict “orientation is well‑defined / not well‑defined” (or use confidence/entropy from the 360‑class distribution) so that slides with missing/ambiguous epidermis can be flagged for manual rotation rather than forcing an uncertain normalization.

3. It may help to clarify that “natural orientation” is typically ill-defined for many non-surface tissues (e.g., lymph node, liver parenchyma) and would require a different definition; otherwise, scope the claim explicitly to skin (and similar surface-epithelium sections).

**Detailed Comments:**

1. Clarify the exact evaluation protocol (synthetic vs. real WSI orientation). Training applies random rotations to already-normalized cross‑sections. Please state explicitly whether the reported test MAEs are computed on:\
(a) original WSI orientations with annotated corrective angles, or\
(b) synthetically rotated normalized images (simulation). \
Figure 3 suggests original WSI orientation is involved, but the text could be interpreted either way. A clear step-by-step description would remove ambiguity.


2. Token subsampling above 15,000 tiles: quantify its effect and ensure determinism at test time. If embeddings are randomly subsampled when >15k tokens, does the same cross‑section yield slightly different predictions across runs? Consider reporting:
* performance as a function of token cap (e.g., 5k/10k/15k), and
* a deterministic inference strategy (fixed sampling or spatially uniform sampling) for deployment.

3. Ablate preprocessing steps that may drive performance. The pipeline includes background replacement, padding correction, and centroid-based position matrices. An ablation would strengthen causal understanding:
* without background replacement,
* without position-matrix-based sinusoidal encoding (or with standard 2D sin/cos grid),
* with/without padding correction rules.

4. The introduction correctly distinguishes “robustness to rotation” from “producing consistent orientation for humans.” Consider explicitly discussing where equivariant models might still matter even though the output is a rotation angle—e.g., as a backbone prior that improves sample efficiency or robustness to artifacts (fragmentation).

5. Explain the “13,082 IHC-stained control sections” in Appendix Fig. A1 and whether they could be used. The flowchart indicates additional IHC control sections. Please clarify what these are (content, label availability) and why excluded. If labels are unavailable, consider whether they could support self‑supervised pretraining or domain adaptation.

6. Unit circle inverse mapping notation needs clarification. The text defines v = ($sin\theta, cos\theta$) = $(x,y)$ and then states $\hat\theta = \mathrm{atan2}(\hat y,\hat x)$. If $x=\sin\theta$ and $y=\cos\theta$, the typical inverse is $\theta=\mathrm{atan2}(x,y)$.  Please confirm the actual implementation to avoid reproducibility issues.

7. If the augmentation is a top-bottom flip, it changes the semantic orientation (epidermis ↔ dermis). If it is a left-right flip, it does not. Please specify which was used and how labels were handled.

8. Since the motivation is viewer integration, report inference time/memory per cross‑section at 1.25×, and expected throughput for a typical case.

9. A small number of near-180° failures can be disproportionately harmful. Reporting the rate of “catastrophic flips” (e.g., |error| > 90°) or 95th percentile absolute error would complement MAE.

**Justification Of The Preliminary Rating:**

The reviewer leans towards weak accept because the paper targets a clear and practical dermatopathology need, includes a large multi‑stain dataset for model development/evaluation, and provides a strong, well-analyzed benchmark showing that 360‑class circular angle classification with wrapped smoothing achieves near-human performance for H&E and competitive results for IHC. The inclusion of inter-/intra‑annotator variability and agreement plots is particularly valuable for judging clinical meaningfulness.

The main factors keeping this from a stronger accept are (i) the lack of an architecture-level sanity check to support the generality of the objective conclusion (which can be addressed via specific rotation-equivariant baselines already cited in the paper), (ii) limited characterization of edge cases where “natural orientation” is absent or ambiguous even within skin, and (iii) missing deployment-centric reporting (runtime/memory and deterministic inference under token caps).

**Questions To Address In The Rebuttal:**

1. Evaluation clarity: Are test results computed on original WSI orientations or synthetic rotations of normalized crops? Please provide a concrete pipeline description.

2. Split integrity across stains: When training on combined H&E+IHC, is the split strictly patient-level across both stains (no patient overlap in any form)?

3. Control sections: What are the 13,082 IHC control sections (Appendix A), and can they be leveraged (supervised or self‑supervised)?

4. Backbone robustness: Can the authors provide at least one additional backbone experiment to show the objective conclusion is not ViT-specific—ideally a rotation-equivariant backbone as in Veeling et al. (2018) or Lafarge et al. (2021), or another transformer variant?

5. Abstention mechanism: Do the authors observe a confidence signal (e.g., entropy/peakiness of the 360‑class distribution) that correlates with failure cases (fragmentation, incomplete epidermis, tangential cuts)? Any preliminary selective-prediction results?

6. Compute: What are inference time and memory requirements per cross‑section, and how does token subsampling affect stability?

7. Implementation detail: Please confirm the correct atan2 mapping used for unit-circle regression and clarify what “vertical flipping” means.

8. All WSIs were analyzed at 1.25x resolution? This means orientation estimation problem does not require the details provided by 20x and 40x scans? This aspect needs to be explicitly explained in the paper.

9. It is not clear how many slides were scanned using a specific scanners in first paragraph on page 3. This needs to be clarified.

10. How many WSIs per patient were included in the training, validation, and testing sets?  How was the aggregation done to report patient level results? Or are the reported results WSI level instead of patient level?

---

> ### Author Response · Authors · 2026-01-25
>
> We thank the reviewer for the detailed review of our manuscript and the suggested improvements. Our response to the suggestions and raised questions are listed below:
>
> Weaknesses:
>
> 1. We fully agree with the reviewer that it is important to investigate if our findings generalize beyond a single deep learning model. For the rebuttal, we considered several different backbone options, but this turned out to be more difficult than initially anticipated. For example, we attempted to train a ResNet18 and EfficientNet-B3, but this resulted in training instability due to the batch normalization operation which is incompatible with our training strategy (i.e., a batch size of 1 with gradient accumulation to allow for cross-section images with large differences in size). We also considered the Swin transformer, but the official implementation does not support training with images of different sizes. The rotation equivariant CNNs mentioned in the introduction are primarily designed to learn invariant representations for downstream tasks like tumor classification, and therefore not suitable for this problem. Hence, we decided to train a smaller Vision Transformer (ViT) configuration (please see Appendix B), which roughly showed the same trends. The table is still missing the result for the orientation classification and currently only reports the results for a single training run for the other approaches, as more was not possible within the one week timeframe of the rebuttal. For the camera ready version of the paper, we would report the complete results (mean and standard deviation) for five training runs.
> 2. We share the opinion of the reviewer that further evaluation of the performance could be valuable. Example cases of some of the failure modes we visually identified are of course already presented in Figure 3. We further investigated the correlation between the absolute error and the size of the cross-section, as well as the number of fragments, but did not observe strong correlations (please see the supporting document). We are unaware of any epidermis segmentation models at low magnification, preventing us to perform further analysis into the association between the epidermis length and the absolute error. Moreover, we included a sentence in the discussion that it would be important to evaluate the model also on non-melanocytic skin pathologies in the future (p. 11). Please see our response to point 5 of the “questions to address in the rebuttal” for the uncertainty estimation analysis.
> 3. In the first paragraph of the introduction, we currently mention: “Unlike tissue sections of most organ types, cross-sections of skin tissue specimens typically have an ideal orientation (i.e., the epidermis positioned at the top to reflect the natural outside-to-inside structure)”, which already informs the reader about the difference in natural orientation between skin tissue cross-sections and sections of other organs. We also express that the scope of our paper is specifically for skin tissue cross-sections: “In this study, we benchmark multiple deep learning approaches for predicting the rotation angle required to correct the misorientation of skin tissue cross-sections.”. While we are open to defining this further, we would like to hear more precisely what is missing in the opinion of the reviewer.
>
> Detailed comments (part 1):
>
> 1. The reported test set results were obtained by evaluating the model predictions on the cross-sections in the test set with their original whole slide image (WSI) orientation (i.e., option (a) in the question). We have clarified this in the manuscript (p. 7).
> 2.  We included an analysis of the predictive performance as a function of the percentage of tokens sampled from the images in Appendix F.
> 3. We included an ablation study to investigate the effect of the positional encodings and background replacement during preprocessing, for which the results can be seen in Table 6. Because the model only allows for input images to have a height and width that is divisible by the patch size, the padding that was performed during preprocessing is necessary and cannot be ablated.
> 4. We have included the improved sample efficiency as a benefit of the rotation equivariant models in the introduction section (p. 2).
> 5. The control sections are tissue sections made from different tissue specimens (sometimes skin but not necessarily, and sometimes but not always as circular cores), which are included on the slides for quality assurance to check whether the IHC staining has been performed correctly. These section do not need to be normalized (and often cannot be normalized) in terms of orientation and were, therefore, excluded. The sections might be useful to include in the future when extending the model with an auxiliary classification head as example cases without a natural orientation.

---

> > ### Author Response · Authors · 2026-01-25
> >
> > Detailed comments (part 2):
> >
> > 6. We thank the reviewer for pointing out this inconsistency. The error was in the definition of x and y. The correct definitions should have been x = cos Θ and y = sin Θ, resulting in Θ = atan2(y, x). We corrected the error in the manuscript (p. 5) and confirmed that the implementation was correct from the start.
> > 7. A left-right flip was used before applying a random rotation during training, which augmented the dataset with mirror images without changing the labels (epidermis remains on top). We have corrected it to “left-right flip” in the manuscript (p. 6).
> > 8. We measured the inference time and the required memory for the cross-section images in the test set, which are reported in Appendix E. We would like to point out that fast inference times and low memory requirements have not been the focus of this study and could likely be further optimized. Moreover, there is often quite a lot of time between the acquisition of the WSI and the moment when a pathologist starts to examine the WSI (e.g., scanning can be done overnight). In a fully digital pathology lab, this provides the opportunity to perform the segmentation and orientation normalization directly after acquisition, such that the model predictions will already be available when the pathologists examine the tissue. Although integrating the model into a WSI viewer when the predictions have not been precomputed would also be useful in certain scenarios, it may not be strictly necessary for routine diagnostic implementation in fully digital pathology labs.
> > 9. We agree with the reviewer that it is important to show how many failure cases there are. For this reason, we included the percentage of images within 2.5°, 5.0°, and 10.0°, which is very similar to the suggested 95th percentile of the mean absolute error. As can be seen in Table 2, roughly 95% of the cases are within 10° of the ground truth for the best performing methods, which means that the 95 percentile of the MAE is around 10°. For the methods with a lower percentage withing 10.0°, the 95 percentile of the MAE will be larger. In addition, for the best performing methods, we also included the Bland-Altman plots, which show the errors for individual data points (i.e., cross-sections). We believe that the combination of these results should give a clear picture on the model performance and the failure cases.
> >
> > Questions To Address In The Rebuttal:
> >
> > 1. Please see our response to point 1 of the detailed comments.
> > 2. Yes, we made sure to use the same patient level split for the H&E and IHC-stained cross-sections in the dataset, meaning that all sections (H&E and IHC) from one patient are either in the train, validation, or test set.
> > 3. Please see our response to point 5 of the detailed comments.
> > 4. Please see our response to point 1 of the weaknesses.
> > 5. We investigated whether the entropy of the predicted probability distribution correlates with the absolute error of the prediction, and if the method would be suitable to detect images of poor quality which were manually removed during the annotation process. The results of this analysis are included in Appendix G.
> > 6. Please see our response to point 8 and 2 of the detailed comments.
> > 7. Please see our response to point 6 and 7 of the detailed comments.
> > 8. Yes, the training and evaluation was strictly performed using cross-sections at 1.25x magnification. Our results, therefore, suggest that high resolution details at 20x and 40x magnification are not necessary to predict the rotation angle for orientation normalization with high accuracy. We hypothesize that the location of the epidermis and specific structures such as hair follicles are most important for the prediction, which can all be discerned at a low magnification. This is also beneficial for computational efficiency, as it does not require processing of many high magnification tiles.
> > 9. In total, 226 WSIs were acquired using the using ScanScope XT scanner, 4,756 WSIs were acquired using the NanoZoomer 2.0-XR scanner, and 2,320 WSIs were acquired using the NanoZoomer S360 scanner. We added this information to the manuscript (p. 3).
> > 10. The data was split at a patient level, meaning that all WSIs from one patient were either in the training, validation, or test set. The number of WSIs per patient slightly differed. For approximately 90% of the patients, only a single H&E-stained WSI and a single corresponding IHC-stained WSI were included. For the remaining 10% of the patients, multiple pairs of H&E and IHC-stained WSIs were included. In some cases, these were from different tissue specimens of the same patient, in and some cases these were from the same specimen. We extracted from all WSIs the cross-sections, and also report all our results at a cross-section level. Hence, no patient-level of WSI-level aggregation was required. We added to the manuscript that the results are reported at a cross-section level (p. 7).

---

### Author Rebuttal · Authors · 2026-01-25

**Rebuttal:**

We thank the reviewers for their feedback on our work. We have attached a revised version of the manuscript that incorporates the suggestions, with all changes highlighted in light blue. In addition, we have included a supplementary document containing three additional analyses that support our responses to several of the points raised below.

**Supporting Material:**

/attachment/9d43644b84637aa9e5f32411ddc465cbfbb32c48.zip

---

### Comment · Area_Chair_vuVb · 2026-02-01
**Please enter Final Rating**

Dear reviewers,
Please note that today, Feb 1 is the last day to enter your final ratings. Thank you to those who have already updated. If you have not yet, please take a moment to look through the author’s rebuttal and update your final score and reasoning.
We greatly appreciate your important contribution to MIDL.
Thank you!
Your AC

---

### Meta-Review · Area_Chair_vuVb · 2026-02-09

**Recommendation:** Accept (Poster)
**Confidence:** 5

**Metareview:**

All reviewers trend toward acceptance of this paper. The greatest concern remaining for some of the reviewers is the motivation/impact of the work. However, reviewers note the work is technically sound and presents interesting results, and the clarifications and additions with the rebuttal were useful. Overall, this paper makes a contribution in the realm of digital pathology for skin tissue analysis.

---

### Decision · Program_Chairs · 2026-02-13

Accept (Poster)